# Exploring the Frontier of Wheat Rust Resistance: Latest Approaches, Mechanisms, and Novel Insights

**DOI:** 10.3390/plants13172502

**Published:** 2024-09-06

**Authors:** Shams ur Rehman, Liang Qiao, Tao Shen, Lei Hua, Hongna Li, Zishan Ahmad, Shisheng Chen

**Affiliations:** 1National Key Laboratory of Wheat Improvement, Peking University Institute of Advanced Agricultural Sciences, Shandong Laboratory of Advanced Agriculture Sciences in Weifang, Weifang 261325, China; shams@pku-iaas.edu.cn (S.u.R.); liang.qiao@pku-iaas.edu.cn (L.Q.); lei.hua@pku-iaas.edu.cn (L.H.); hongna.li@pku-iaas.edu.cn (H.L.); 2State Key Laboratory of Tree Genetics and Breeding, Co-Innovation Centre for Sustainable Forestry in Southern China, Bamboo Research Institute, Key Laboratory of National Forestry and Grassland Administration on Subtropical Forest Biodiversity Conservation, School of Life Sciences, Nanjing Forestry University, Nanjing 210037, China; ahmad.lycos@gmail.com

**Keywords:** wheat rusts, genetic resistance, pathogen epidemiology, molecular mechanisms, disease management, climate change impacts

## Abstract

Wheat rusts, including leaf, stripe, and stem rust, have been a threat to global food security due to their devastating impact on wheat yields. In recent years, significant strides have been made in understanding wheat rusts, focusing on disease spread mechanisms, the discovery of new host resistance genes, and the molecular basis of rust pathogenesis. This review summarizes the latest approaches and studies in wheat rust research that provide a comprehensive understanding of disease mechanisms and new insights into control strategies. Recent advances in genetic resistance using modern genomics techniques, as well as molecular mechanisms of rust pathogenesis and host resistance, are discussed. In addition, innovative management strategies, including the use of fungicides and biological control agents, are reviewed, highlighting their role in combating wheat rust. This review also emphasizes the impact of climate change on rust epidemiology and underscores the importance of developing resistant wheat varieties along with adaptive management practices. Finally, gaps in knowledge are identified and suggestions for future research are made. This review aims to inform researchers, agronomists, and policy makers, and to contribute to the development of more effective and sustainable wheat rust control strategies.

## 1. Introduction

According to FAO (https://www.fao.org/faostat/en/#data, accessed on 8 July 2024), wheat stands as one of the paramount cultivated food crops worldwide. It accounts for approximately 20% of the global human calorie intake. Each year, wheat cultivation spans across approximately 219 million hectares, and yields over 760 million tons. In 2023, around 136.6 million metric tons of wheat were produced in China, making the country the world’s leading wheat producer. USA is the fourth largest producer and the second largest exporter of wheat. Pests and diseases cause 20–40% of global food crop losses and losses of USD 220 billion in agricultural trade every year. Wheat rusts, comprising leaf, stem, and stripe rusts, are significant problems for farmers around the world, causing major yield losses in one of the world’s most important staple crops [1]. The history of wheat rusts dates back centuries, with records of severe epidemics that drastically reduced wheat yields and led to food shortages [2]. The economic implications are profound, particularly in countries where wheat is a major agricultural product [3]. Wheat rust has a dynamic nature; it continually evolves, overcoming resistance in wheat varieties, challenging the effectiveness of traditional breeding methods and necessitating ongoing research and adaptation [4]. Understanding wheat rust’s background is crucial as it offers insights into the pathogen’s evolution, interaction with host plants, and development of resistance over time [5]. The significance of wheat rust research extends beyond agricultural economics, encompassing ecological and environmental concerns, particularly due to the risks posed by widespread fungicide use, which can impact environmental health and sustainability [6]. Assessing these ecological impacts is essential for developing integrated and sustainable management strategies [7].

This review offers a comprehensive yet focused examination of wheat rust, covering its etiology, epidemiology, and global distribution. It emphasizes advances in genetic resistance through traditional breeding and modern genomic approaches, which have been pivotal in combating the pathogen. The review also explores the molecular mechanisms underlying pathogenesis and host–pathogen interactions, essential for developing effective control strategies and resistant wheat varieties. Additionally, it discusses innovative management strategies, including fungicides, biological control, and integrated pest management. The impact of climate change on wheat rust is highlighted, stressing the importance of adaptive strategies and resilient varieties. Finally, the review identifies future research directions, pinpointing gaps in current knowledge and opportunities for innovation and collaboration.

This review offers a comprehensive synthesis of recent wheat rust research, emphasizing advances in genomics, breeding techniques, and management strategies. It underscores the impact of climate change on disease outbreaks, urging the integration of eco-evolutionary theories and science–policy interface for effective management. Bridging traditional agriculture with modern science, it serves as a vital resource for researchers, policy makers, and practitioners in the field of wheat rust management, aiming to address challenges, promote sustainability, and guide future research directions in wheat rust management.

## 2. Etiology and Epidemiology of Wheat Rust

Wheat rusts are caused by pathogenic fungi that belong to the genus *Puccinia*, which are obligate parasites, meaning they require a living host to complete their life cycle [8]. The three main types of wheat rust diseases, leaf (brown) rust, stem (black) rust, and stripe (yellow) rust, are caused by distinct species of *Puccinia* genus: *Puccinia triticina*, *Puccinia graminis* f. sp. *tritici*, and *Puccinia striiformis* f. sp. *tritici*, respectively. Each type affects different parts of the wheat plant and has unique characteristics [9].

### 2.1. Types of Wheat Rust Diseases: Leaf Rust, Stem Rust, and Stripe Rust

The most common types of wheat rust continue to pose significant threats to wheat production globally, with recent studies highlighting new challenges and advances in resistance breeding. Leaf rust, caused by *P. triticina*, remains widespread and problematic, especially in regions with cooler, wetter conditions. The fungus adapts rapidly to resistant wheat varieties, leading to new virulent races that overcome existing resistance [10]. Recent studies emphasize the importance of understanding the molecular interactions between the pathogen and wheat, including both pre-haustorial and post-haustorial resistance mechanisms [11,12]. Notably, systemic acquired resistance has been highlighted as a promising method for providing durable protection against a range of pathogens [13,14,15].

Stem rust, caused by *P. graminis*, has historically been one of the most devastating wheat diseases [16]. Although less common today due to effective resistant varieties, it remains a significant threat, particularly in regions with warmer climates [17]. Recent research has focused on identifying and mapping rust resistance loci across wheat genomes, aiding in the development of new, more resilient wheat varieties [16,18,19].

Stripe rust, caused by *P. striiformis*, thrives in cooler climates and rapidly spreads under favorable conditions [20]. It is characterized by yellow or orange stripe-like pustules on the leaves and can significantly reduce yield quality and quantity. Recent outbreaks have been reported even in regions that previously experienced minimal issues, emphasizing the pathogen’s adaptability [21,22,23]. Effective management strategies include the use of fungicides and the cultivation of resistant varieties, although the constant evolution of the pathogen requires ongoing monitoring and adaptation [7,20,24,25,26].

### 2.2. Global Distribution and Economic Impact

The epidemiology of wheat stem rust is influenced by climatic conditions. It thrives in high temperatures ranging from 25 to 30 °C during the day and milder conditions at night, around 15–20 °C. Elevated relative humidity is also conducive to the development of stem rust. Infection typically takes place through the plant’s natural openings, such as the stomata [27]. These diseases are found in major wheat-growing areas across the world, including North America, Europe, Asia, Africa, and Australia [28]. A disease spatial distribution map of wheat rusts (major: severe loss, minor: occur less significantly, local: occur at smaller scale, rare: rarely observed) is illustrated in Table 1 [29]. Wheat rusts have a profound economic impact on wheat production, with each type of rust causing varying degrees of yield loss. Stem rust (*P. graminis*), one of the most devastating forms, can result in yield losses ranging from 50% to 100% in severe cases, particularly when highly susceptible varieties are affected [30]. Leaf rust (*P. triticina*), while generally less severe than stem rust, can still lead to significant losses, typically ranging from 10% to 40% depending on the severity of the outbreak and the resistance of the wheat variety [31]. Stripe rust (*P. striiformis*), which thrives in cooler climates, can cause yield losses of 10% to 70%, especially under epidemic conditions or when wheat varieties lack resistance [32]. However, in cases of severe stem rust epidemics, particularly with highly susceptible wheat varieties and under optimal conditions for the pathogen, yield losses can reach up to 100%, resulting in total crop failure [30,31,33]. The financial cost includes lost yield and an increased need for fungicides and the development of resistant wheat varieties [34].

The continuous evolution of rust pathogens, overcoming resistant wheat varieties, poses a constant challenge to global food security. The Ug99 stem rust strain (race TTKSK), first identified in Uganda in 1999, is an example of a highly virulent strain that has spread to several countries and overcome many previously resistant wheat varieties [35]. The ongoing emergence and spread of highly virulent rust pathogens, such as the Ug99 strain of stem rust, jeopardize wheat crops worldwide by breaching the defenses of resistant varieties [36,37]. In conclusion, understanding the etiology and epidemiology of wheat rust is crucial for developing effective control and management strategies. Continuous research and global collaboration are required to monitor the spread of these diseases and to breed resistant wheat varieties, ensuring the stability of wheat production worldwide.

## 3. Advances in Genetic Resistance to Wheat Rusts

### 3.1. Traditional Breeding Techniques

The battle against wheat rusts has historically relied on traditional breeding techniques. These methods involve selecting wheat varieties that show natural resistance to rust and cross-breeding them with high-yielding but susceptible varieties [38]. This approach has been fundamental in developing wheat varieties that combine desirable agronomic traits with rust resistance [39]. Plant breeders have employed various strategies such as backcrossing, where a resistant variety is repeatedly crossed with a high-yielding variety to combine resistance with other favorable traits. Another method is the multiline approach, where several genetically different lines, all resistant to rust, are planted together. This strategy reduces the likelihood of rust pathogens overcoming the plant’s resistance [40]. Traditional breeding techniques have played a pivotal role in the battle against wheat rusts, fostering the development of wheat varieties that not only exhibit natural resistance to rust but also possess desirable agronomic traits, ensuring both productivity and resilience in the face of disease challenges. By understanding how these techniques contribute to wheat’s resilience and yield, the information provides valuable insights for future breeding efforts aimed at ensuring the long-term sustainability and health of wheat crops.

### 3.2. Genomic Approaches and Genetic Engineering

The advent of genomic approaches has revolutionized wheat breeding for rust resistance. Genomic techniques, such as marker-assisted selection (MAS), allow breeders to identify and select for resistance genes more efficiently and accurately [41]. MAS uses molecular markers that are linked to resistance genes, speeding up the breeding process by eliminating the need for extensive field testing [42].

Historically, the cloning of resistance genes and other genes of interest predominantly relied on a map-based cloning strategy, a method necessitating the use of genetic and physical maps, which was both laborious and time-intensive. However, fine mapping around recombination suppression regions, where many alien resistance genes are localized, posed significant challenges. In contrast, DNA and/or RNA sequencing-based techniques have emerged as alternative methodologies, employing independent mutants to isolate candidate genes. These approaches target specific chromosomes or gene classes, circumventing genome-wide interference and obviating the need for a genetic map. Notably, recent advancements have led to the cloning of several rust-resistance genes utilizing sequencing-based methodologies. For instance, *Yr7*, *Yr5/YrSP*, *Sr9* [43], *Sr22*, *Sr26*, *Sr45*, and *Sr61* were successfully cloned via MutRenSeq [43,44,45,46]; *Sr43* was elucidated utilizing the MutChromSeq technique [47,48]; and *Sr62* was identified through MutRNA-Seq [49]. Nonetheless, it is essential to acknowledge that each of these methods harbors distinct limitations. For instance, MutRenSeq is tailored specifically for nucleotide binding and leucine-rich repeat (NLR) gene cloning; MutChromSeq entails a technically demanding step of chromosome sorting via flow cytometry; and MutRNA-Seq necessitates a high-quality genome assembly of the wild-type (WT) genotype. Approximately 18 stem rust (*Sr*), ten yellow rust (*Yr*), and eleven leaf rust (*Lr*) resistance genes have been cloned and characterized [43,50,51].

Figure 1 illustrates various molecular approaches for wheat rust, detailing methods used to detect and incorporate resistance genes into new wheat varieties. The third-generation hybrid wheat technology leverages advanced genetic tools to produce superior wheat hybrids with enhanced rust resistance [52]. The application of haploid technology can significantly accelerate the breeding process in wheat breeding by quickly producing plants that are homozygous for desired traits, including resistance to various rust pathogens [4]. Genetic engineering, another groundbreaking approach, involves directly inserting specific resistance genes into the wheat genome. This method has enabled the introduction of novel resistance genes that are not naturally present in wheat [53]. For instance, genes conferring resistance to stem rust have been successfully transferred from related species into wheat, offering new avenues for creating rust-resistant varieties. Three genes, *Sr33*, *Sr45*, and *Sr46*, which confer resistance to stem rust, were originally identified in *Aegilops tauschii*. These genes have been successfully transferred to wheat through traditional breeding methods, specifically using chromosome engineering and backcrossing techniques. These methods allow for the incorporation of desirable traits from wild relatives like *Aegilops tauschii* into cultivated wheat, thereby enhancing resistance to stem rust [54]. Furthermore, an efficient CRISPR-Cas9 multiplex system had successfully been established for simultaneous genome editing at multiple loci in wheat, which greatly shortened the breeding time [55]. Using CRISPR/Cas9, negative regulatory genes have been edited to enhance wheat yields and quality. For instance, knocking out GASR7, a gibberellin-regulated gene controlling grain length in rice, and targeting all three TaGASR7 homoeologs simultaneously led to a significant increase in thousand kernel weight across different wheat varieties [56]. Similarly, editing GW2, which encodes a RING-type E3 ligase controlling grain weight in rice, resulted in the enlargement of wheat grains, thus boosting grain yields [57,58].

The integration of genomic approaches holds great potential to revolutionize wheat breeding strategies for rust resistance, offering expedited methods such as marker-assisted selection and third-generation hybrid wheat technology. However, this potential has only been realized in a few very specific cases, and widespread implementation remains limited. The promise of these technologies is significant, but more research and development are required before they can be broadly applied across different wheat varieties and environments. Molecular tools like haploid technology and genetic engineering have further enhanced breeding efficiency and expanded genetic diversity. While the establishment of an efficient CRISPR-Cas9 multiplex system holds significant promise for shortening breeding times and enhancing wheat yields and quality, this potential has only been realized in a few specific cases. The technology is still in the early stages of application, and although it offers considerable possibilities, widespread implementation and consistent results across different wheat varieties have yet to be fully achieved [59,60,61]. These advancements underscore the transformative potential of genomics in addressing rust resistance challenges and advancing sustainable wheat agriculture.

### 3.3. Identifying Key Resistance Genes

Identifying key resistance genes and understanding the underlying mechanisms are crucial for developing durable rust-resistant wheat varieties. Resistance genes in plants, commonly referred to as R genes, endow them with the capability to identify and mount a defense against rust pathogens. However, it should be noted that not all resistance genes fall under the category of R genes. Several methods have been developed for the rapid cloning of resistance genes (R genes) involved in wheat rust. One such method is the map-based cloning approach, which involves the construction of high-density genetic maps using molecular markers linked to R genes. This method facilitates the identification and isolation of candidate R genes by positional cloning techniques. Additionally, next-generation sequencing (NGS) technologies have accelerated the cloning process by enabling whole-genome sequencing of wheat and its wild relatives, allowing for the identification of R genes through comparative genomics approaches [42]. Moreover, advanced molecular techniques such as MutChromSeq (Mutagenesis Chromosome Sequencing) have been employed to rapidly identify candidate R genes by sequencing specific chromosomal regions harboring mutations conferring resistance to rust pathogens [62,63].

Extensive research has led to the identification of several R genes effective against different types of wheat rust [64]. One of the most significant breakthroughs in this area is the identification of the *Sr31* gene, which provided resistance to stem rust for many years before the emergence of the Ug99 strain. The discovery of the *Sr31* gene marked a major advancement in plant pathology, resisting stem rust for an extended period until it was overcome by the Ug99 strain. Ongoing research efforts have led to the mapping and cloning of additional R genes like *Sr35* and *Sr50*, which are effective against Ug99 and other aggressive strains. To provide a more comprehensive overview, further examples, particularly those pertaining to leaf rust and stripe rust resistance genes, would enhance the understanding of genetic resistance across different rust diseases [65]. A list of cloned genes with wheat rust resistance is presented in Table 2 [66]. The challenge lies in the rust pathogen’s ability to evolve and overcome single R gene resistance. To address this, researchers are focusing on stacking multiple R genes in a single wheat variety, a strategy known as gene pyramiding. This approach aims to provide a more durable and broad-spectrum resistance to wheat rusts [67]. By employing advanced genomic techniques and gene pyramiding strategies, the enhancement, effectiveness, and durability of rust resistance in wheat can be achieved. This approach, by mitigating the impact of evolving rust pathogens and ensuring stable yields, is crucial for improving wheat crop sustainability. Challenges include the complexity of rust pathogen diversity, regulatory hurdles in deploying genetically modified crops, and the need for sustainable management practices to maintain long-term resistance in wheat varieties. Addressing these challenges will be essential for securing global wheat production against evolving rust threats. In addition, developing innovative strategies like gene editing and genomic selection to counteract it will enhance the understanding of rust pathogen evolution.

## 4. Molecular Mechanisms of Wheat Rust Pathogenesis

### 4.1. Pathogen Life Cycle and Infection Process

The life cycle of wheat rust pathogens, namely *Puccinia* species, is complex and involves several stages. These fungi are heteroecious, requiring two different host species to complete their life cycle: a primary host (wheat) and an alternate host (*Berberis vulgaris* L.), as explained in Figure 2. The life cycle includes both sexual and asexual stages, which contribute to the genetic diversity and adaptability of these pathogens [35]. The infection process begins when rust spores land on the wheat leaf surface. Under favorable conditions (such as moisture and appropriate temperature), the spores germinate and produce specialized structures called appressoria. These penetrate the plant’s cuticle and epidermal cells. Once inside, the pathogen forms haustoria, which are structures that extract nutrients from the host cells, allowing the rust to grow and proliferate within the plant tissue [34].

### 4.2. Host–Pathogen Interactions

The interaction between wheat and rust pathogens is a complex process involving both plant defenses and pathogen strategies. Wheat detects pathogen-associated molecular patterns (PAMPs) via pattern recognition receptors (PRRs), initiating PAMP-triggered immunity (PTI) [77]. Rust fungi counteract this by secreting effectors that suppress PTI, leading to effector-triggered susceptibility (ETS) [78]. Wheat, in turn, has evolved resistance (R) genes that activate a stronger defense, known as effector-triggered immunity (ETI) [14]. Key components in wheat–rust interactions include PRRs that trigger initial defenses and nucleotide-binding leucine-rich repeat (NLR) proteins that play a critical role in ETI by recognizing fungal effectors and triggering the hypersensitive response (HR) [79]. Additionally, small RNAs, epigenetic changes, and hormonal signaling regulate these defenses [80].

Advances in genomics have facilitated the development of rust-resistant wheat varieties through the cloning and deployment of R genes [48,81]. However, the rapid evolution of rust pathogens necessitates the stacking of multiple resistance genes for durable protection [51]. Regulatory constraints and societal acceptance of genetically modified crops also pose challenges to deploying genetically engineered solutions [50].

Continued research in genomics and biotechnology is crucial for enhancing wheat’s resilience against rust pathogens and ensuring global food security [82]. Further efforts to unravel the genetic basis of wheat resistance to rust pathogens, integrate omics technologies to identify novel resistance genes, and leverage genome editing for precise modification of plant immunity are crucial. Enhanced understanding of small RNA-mediated regulation, epigenetic modifications, and signaling pathways will enable more targeted approaches to boost wheat resilience against evolving rust threats [51]. The application of advanced biotechnological tools in breeding programs holds promise for developing climate-resilient wheat varieties with robust and lasting resistance to rust diseases, ensuring global food security amidst changing environmental conditions [83].

### 4.3. Signal Transduction and Defense Mechanisms

Upon recognition of rust pathogens, a complex signal transduction network is activated in wheat plants. This network involves various signaling molecules, such as salicylic acid (SA), jasmonic acid (JA), and ethylene (ET), which play crucial roles in modulating the plant’s defense response [84]. Salicylic acid is particularly important in the defense against biotrophic pathogens like wheat rust. It activates the expression of pathogenesis-related (*PR*) genes, which encode proteins that contribute to plant defense, such as chitinases and glucanases [85]. Significant progress has been made in elucidating the signal transduction pathways and defense mechanisms associated with key resistance genes in wheat, including *Yr36*, *Lr67*, and *Lr34*. *Yr36* is a novel kinase-START protein that confers resistance to yellow rust caused by *P. striiformis* f. sp. *tritici*. Mechanistic studies have revealed that Yr36 functions as a receptor-like kinase (RLK) that recognizes specific effector proteins secreted by the pathogen, triggering downstream defense responses. Activation of mitogen-activated protein kinase (MAPK) cascades, calcium signaling, and reactive oxygen species (ROS) production are among the key events implicated in the *Yr10*-mediated defense response [86].

Similarly, *Lr67* and *Lr34* are well-characterized resistance genes providing durable resistance against leaf rust caused by *P. triticina*. *Lr67* encodes a hexose transporter-like protein and is associated with broad-spectrum resistance. Recent studies have demonstrated that *Lr67*-mediated resistance involves the modulation of defense-related genes, including those associated with salicylic acid (SA) signaling pathways and cell wall reinforcement [87]. On the other hand, *Lr34* encodes an ATP-binding cassette (ABC) transporter and is known for its adult plant resistance (APR) phenotype. Research on *Lr34* has highlighted its role in enhancing basal defense responses and priming systemic acquired resistance (SAR) through the upregulation of defense-related genes and phytohormone signaling pathways [88]. Overall, mechanistic studies of *Yr36*, *Lr67*, and *Lr34* have provided valuable insights into the signal transduction pathways and defense mechanisms underlying wheat rust resistance, contributing to the development of effective strategies for disease management and crop improvement. Furthermore, reactive oxygen species (ROS) are produced as a part of the defense response, creating a hostile environment for the pathogen and signaling for the activation of further defense mechanisms [89]. In addition to these molecular responses, structural changes, such as the reinforcement of the cell wall at the site of infection, also occur. This lignification of cell walls helps to restrict the spread of the pathogen [90]. In conclusion, understanding the molecular mechanisms of wheat rust pathogenesis is critical for developing effective control strategies. This includes breeding for resistance, designing targeted fungicides, and developing integrated pest management strategies.

However, the genetic variability of rust pathogens can be one of the limitations which can lead to the breakdown of resistance conferred by single genes over time. Additionally, the deployment of genetically modified crops and resistance genes faces regulatory hurdles and societal acceptance issues. Further research is needed to fully understand the complex interactions between wheat and rust pathogens and to identify additional resistance mechanisms beyond those currently characterized. Future efforts will focus on leveraging genomic tools and biotechnological advancements to identify and deploy novel resistance genes in wheat breeding programs. Techniques like genome editing and RNA interference hold promise for enhancing the durability and specificity of resistance against diverse rust pathogens. Ultimately, enhancing wheat rust resistance contributes to global food security by reducing yield losses and ensuring stable production in the face of evolving pathogen pressures.

## 5. Innovative Management Strategies for Wheat Rust

### 5.1. Developments in Fungicides

Advancements in fungicide development have played a crucial role in managing wheat rust. Modern fungicides are more targeted and effective, with a lower environmental impact than their predecessors [91]. New formulations and active ingredients have been developed to target specific stages of the rust pathogen’s life cycle, reducing the likelihood of resistance development [92]. One significant development is the use of strobilurins and triazoles, which inhibit different pathways in the fungal cells, ensuring effective control of rust. Additionally, there has been a move towards systemic fungicides, which are absorbed by the plant and provide protection over a longer period [93]. Advancements in fungicide development for managing wheat rust have led to more targeted and effective treatments with reduced environmental impact (Table 3). Innovative formulations and active ingredients, such as strobilurins and triazoles, inhibit specific fungal pathways, while systemic fungicides offer prolonged protection against rust pathogens.

The development of advanced fungicides for wheat rust management addresses the urgent need for effective disease control to safeguard global wheat production. These innovations not only improve crop yields by reducing disease impact but also contribute to food security by ensuring stable wheat supplies. Despite advancements, challenges in fungicide development for wheat rust management include the potential for resistance development in rust pathogens due to prolonged and widespread fungicide use. Additionally, environmental concerns persist regarding the ecological impact of fungicides, particularly systemic ones that can accumulate in the environment. Regulatory hurdles and the cost-effectiveness of newer formulations also pose challenges, limiting accessibility for farmers in resource-constrained regions. Future research should focus on developing sustainable fungicide strategies that minimize environmental impact while maximizing efficacy against evolving rust pathogen strains. This includes exploring novel modes of action, synergistic combinations of active ingredients, and formulations that enhance adherence and persistence on plant surfaces. Understanding the genetic basis of fungicide resistance in rust pathogens and developing rapid diagnostics for monitoring resistance emergence are critical for sustainable disease management.

### 5.2. Biological Control and Integrated Disease Management

Biological control offers an eco-friendly approach to managing wheat rust by utilizing natural antagonists or enemies of the rust pathogen. This method is attractive due to its sustainability and minimal environmental impact. Examples include using fungi or bacteria that are antagonistic to rust pathogens or that enhance the plant’s natural defense mechanisms [6]. Integrated Disease Management (IDM) builds on biological control by combining it with other strategies such as crop rotation, the use of resistant wheat varieties, and judicious application of fungicides. IDM aims to manage wheat rust holistically, reducing chemical reliance while promoting sustainable agricultural practices [100,101].

Despite the promise of biological control, its application faces several challenges. Environmental variability can influence the effectiveness of natural antagonists, necessitating precise timing and favorable conditions for optimal results. Additionally, compatibility issues may arise between biological control agents and other disease management practices, such as fungicide application. Ongoing research to discover novel biological control agents and optimize their application methods is crucial to improving our understanding of the interactions between natural antagonists and rust pathogens under diverse environmental conditions. Long-term studies assessing the ecological impacts and economic feasibility of incorporating biological control into IDM strategies are essential to enhancing the overall effectiveness of wheat rust management [102,103,104].

The *Lr34* gene has been widely utilized not only for its broad-spectrum resistance to multiple rust diseases, including leaf rust, stripe rust, and stem rust, but also for its associated traits that contribute to enhanced tolerance to environmental stresses like drought and heat [105]. The incorporation of *Lr34* into wheat varieties has proven effective in mitigating the combined pressures of rust pathogens and climate-induced stresses, contributing to more resilient crop performance under adverse conditions, as illustrated in Figure 3 [104,106].

Managing wheat rust through resistance breeding has evolved significantly, with modern efforts emphasizing the development of varieties that combine multiple resistance genes, such as *Yr18*, *Sr26*, and *Lr67*, to provide durable protection against diverse rust strains [107]. Advances in genome editing technologies, particularly CRISPR-Cas9, have revolutionized the precision with which resistance genes can be introduced or modified within wheat genomes, offering a faster and more targeted approach than traditional methods [108]. Marker-assisted selection (MAS) has further refined breeding programs by streamlining the identification and introgression of resistance genes into elite wheat lines, ensuring high yield potential without sacrificing disease resistance [109]. Additionally, the application of third-generation hybrid wheat technology and haploid induction techniques has accelerated the breeding process [4]. These methods allow for the rapid development of pure lines and hybrids that exhibit enhanced resistance and overall crop resilience under both biotic and abiotic stresses [110]. The integration of these advanced tools and technologies into a broader strategy that includes fungicide use, biological control, and integrated disease management is crucial for mitigating the impact of wheat rust on global production, particularly in the face of climate change and evolving pathogen pressures.

## 6. Impact of Climate Change on Wheat Rust

### 6.1. Changes in Wheat Rust Epidemiology

Climate change has a significant impact on the epidemiology of wheat rust diseases (Figure 4). Rising temperatures, altered rainfall patterns, and increased extreme weather events can influence the distribution, severity, and timing of wheat rust outbreaks [111]. Warmer temperatures may extend the growing season of the wheat crop, but they also favor the development and spread of rust pathogens, potentially leading to more frequent and severe infections [112]. Changes in humidity and rainfall can affect spore germination and dispersal. For example, increased humidity can promote rust development, while changes in wind patterns can facilitate the spread of rust spores over longer distances [113]. Additionally, the stress imposed on plants by changing environmental conditions can reduce their natural resistance to diseases, making them more susceptible to rust infections [114]. Climate change exacerbates wheat rust diseases by altering environmental conditions, favoring pathogen development and spread, thereby increasing the frequency and severity of infections.

### 6.2. Adapting Wheat Cultivation to Climate Change

Adapting wheat cultivation practices to climate change is essential for managing rust outbreaks and ensuring crop productivity. Strategies include adjusting sowing dates to minimize exposure during peak rust seasons, employing crop rotation and diversification to lower pathogen loads, and implementing conservation agriculture techniques to enhance soil health and plant resilience [115]. Additionally, the use of precision agriculture technologies, such as remote sensing and predictive modeling, is being explored to monitor rust development in real-time. These technologies offer the potential for early detection of rust outbreaks, enabling timely and targeted interventions to protect crops and maintain yield quality [116].

### 6.3. Developing Climate-Resilient Wheat Varieties

Breeding and developing climate-resilient wheat varieties is a critical strategy to combat the impact of climate change on wheat rust. This involves not only enhancing resistance to rust diseases but also improving the overall resilience of wheat plants to environmental stressors such as drought, heat, and salinity [117]. In conclusion, climate change poses significant challenges to managing wheat rust diseases. Addressing these challenges requires a comprehensive approach that includes understanding and monitoring the changing epidemiology of rust, adapting cultivation practices, and developing wheat varieties that are resilient to both rust diseases and the broader impacts of climate change. Furthermore, developing climate-resilient wheat varieties involves complex genetic interactions and regulatory frameworks that govern the deployment of genetically modified crops, posing challenges related to public acceptance and regulatory approval. The adaptation of wheat cultivation practices requires substantial changes in agronomic practices and farmer behavior, which may face resistance due to economic constraints or traditional farming practices. Implementing precision agriculture technologies for monitoring and early detection of rust outbreaks requires significant investment in infrastructure and training, particularly in developing regions where resources may be limited.

## 7. Future Directions and Research Gaps

### 7.1. Unexplored Areas in Wheat Rust Research

Future research in wheat rust management should prioritize specific and impactful areas that can significantly advance our understanding and control of the disease. One promising but underexplored avenue is the interaction between wheat rust pathogens and the plant’s microbiome. Investigating how beneficial microbes can enhance wheat’s innate resistance to rust could reveal new, sustainable disease control methods [82]. Additionally, understanding the long-term evolutionary dynamics between wheat and rust pathogens is critical for predicting future changes in pathogen virulence and developing durable resistance strategies. This can be achieved through longitudinal studies and evolutionary modelling to monitor changes in pathogen populations [118].

Recent advancements have also seen the successful incorporation of resistance genes from wild relatives of wheat, such as Lr47 and Lr42, into modern wheat varieties. These genes offer broad-spectrum resistance to leaf rust and were integrated using advanced gene-editing techniques like CRISPR-Cas9, as well as traditional breeding methods. Future research should focus on stacking these genes with other resistance loci and testing their effectiveness across diverse environmental conditions to ensure their durability [51]. The application of functional genomics and genome-wide association studies (GWASs) has been instrumental in uncovering new sources of resistance. For example, a GWAS on elite durum wheat has identified significant loci like Yrdurum-1BS.1 for stripe rust resistance, while other studies have pinpointed multiple genomic regions linked to resistance against various rust diseases. Continued expansion of GWASs and meta-QTL analysis will be crucial for integrating these findings into marker-assisted selection (MAS) and broadening the genetic base of rust-resistant wheat [119].

Moreover, leveraging high-throughput omics technologies such as transcriptomics and proteomics can provide deeper insights into the molecular mechanisms governing host–pathogen interactions in rust infections. These insights can be used to identify key regulatory genes and pathways involved in rust resistance, facilitating the development of new resistant cultivars [120,121]. Finally, as climate change continues to impact agricultural systems, it is vital to understand its influence on the epidemiology of rust diseases. Combining climate modelling with rust epidemiology will enable the development of predictive tools that guide planting schedules, resistance breeding, and disease management practices, ensuring that wheat cultivation remains resilient to environmental changes [25,122].

### 7.2. Potential for Technological Innovations

The field of wheat rust research holds substantial potential for technological innovations. Advances in genomics, including next-generation sequencing and CRISPR-Cas9 gene-editing technology, offer promising tools for dissecting the genetic basis of rust resistance and for developing new resistant wheat varieties. The field of genomics, particularly propelled by next-generation sequencing (NGS) technologies, has revolutionized our ability to understand the genetic intricacies underlying rust resistance in wheat. NGS facilitates high-throughput sequencing of entire genomes, enabling researchers to decipher the genetic makeup of wheat varieties with varying degrees of rust resistance. Moreover, the advent of CRISPR-Cas9 gene-editing technology has further augmented our capacity to precisely manipulate the wheat genome, facilitating the targeted modification of genes associated with rust resistance. With CRISPR-Cas9, researchers can efficiently introduce beneficial genetic variations or eliminate detrimental ones, thereby accelerating the development of rust-resistant wheat varieties [82,123]. For example, they successfully knocked out the *TaPsIPK1* gene in wheat, conferring broad-spectrum resistance to the stripe rust pathogen *Pst* without compromising important agronomic traits in field tests [124]. The integration of these cutting-edge genomic tools into wheat breeding programs holds significant promise for developing durable rust-resistant wheat varieties. By precisely identifying and manipulating genes associated with rust resistance, breeders can expedite the development of cultivars with enhanced resistance to diverse rust pathogens.

Remote sensing technologies, coupled with artificial intelligence (AI)-driven predictive modeling, offer unprecedented capabilities for the real-time monitoring and management of rust outbreaks in agricultural landscapes. These innovative approaches enable the rapid detection and assessment of rust infestations over large geographical areas, providing valuable insights into disease dynamics and spatial distribution patterns. For instance, satellite imagery and unmanned aerial vehicles (UAVs) equipped with advanced sensors can capture detailed information about crop health indicators, such as chlorophyll content and canopy temperature, which are closely associated with rust infection levels [125]. Furthermore, AI algorithms trained on these remote sensing data can analyze complex patterns and predict the likelihood of rust outbreaks with high accuracy and efficiency [126]. By integrating remote sensing data with weather forecasts and crop growth models, decision-support systems can deliver timely recommendations for targeted intervention strategies, including optimal timing and dosage of fungicide applications [127].

These technologies can help in early detection and in making informed decisions on fungicide application and other management practices. Additionally, there is potential in exploring novel fungicides and biological control agents that are more effective, sustainable, and environmentally friendly.

### 7.3. Policy and Research Recommendations

To effectively address the challenges posed by wheat rust, a coordinated approach involving policy and research is essential. Policies that promote research and development in rust-resistant wheat breeding, sustainable farming practices, and climate-smart agriculture are crucial. Investment in research infrastructure, including disease monitoring networks and genetic research facilities, is vital for advancing our understanding and management of wheat rust. Collaborations between governments, research institutions, and the agricultural industry are necessary to foster innovation and the rapid deployment of new technologies and strategies. Furthermore, there is a need for international cooperation in research and data sharing, given the global nature of wheat rust. Policies that encourage such collaboration can significantly enhance the effectiveness of rust management strategies worldwide. In conclusion, while significant progress has been made in wheat rust research, there are still numerous opportunities for further exploration and innovation. A concerted effort involving technological advancement, policy support, and international collaboration is key to addressing the ongoing challenges posed by wheat rust.

## 8. Conclusions

This discussion highlights the crucial role of combining traditional breeding and modern genomic techniques in developing rust-resistant wheat. Advances in fungicides, biological control, and integrated disease management offer promising strategies against wheat rust. Additionally, this review underscores the importance of resilient and adaptable wheat varieties in response to the challenges posed by climate change on wheat rust epidemiology. The findings from this review offer new insights and perspectives, incorporating recent advancements in wheat rust research that can significantly inform both academic research and practical applications in agriculture. For researchers, the identified gaps offer opportunities to explore the complex interactions between wheat, rust pathogens, and environmental factors. For practitioners in agriculture and plant breeding, this review provides insights into sustainable rust management strategies. Adopting advanced breeding techniques, integrated management practices, and technology for monitoring and early detection can significantly improve rust control efforts.

Wheat rust remains a significant agricultural challenge, but ongoing research and technological advancements offer hope. An integrated approach, combining scientific innovation, sustainable practices, and sound policy, is essential to combat this threat. Global collaboration and information sharing are crucial, as wheat rust is a worldwide issue requiring a unified response. This report underscores the importance of continued research and innovation to protect global food security. Training the next generation of plant pathologists and agronomists, strengthening surveillance infrastructure, and raising public awareness are vital steps. Lastly, partnerships between governments, research institutions, and farmers are key to developing and implementing effective, practical strategies on a global scale.

## Figures and Tables

**Figure 1 plants-13-02502-f001:**
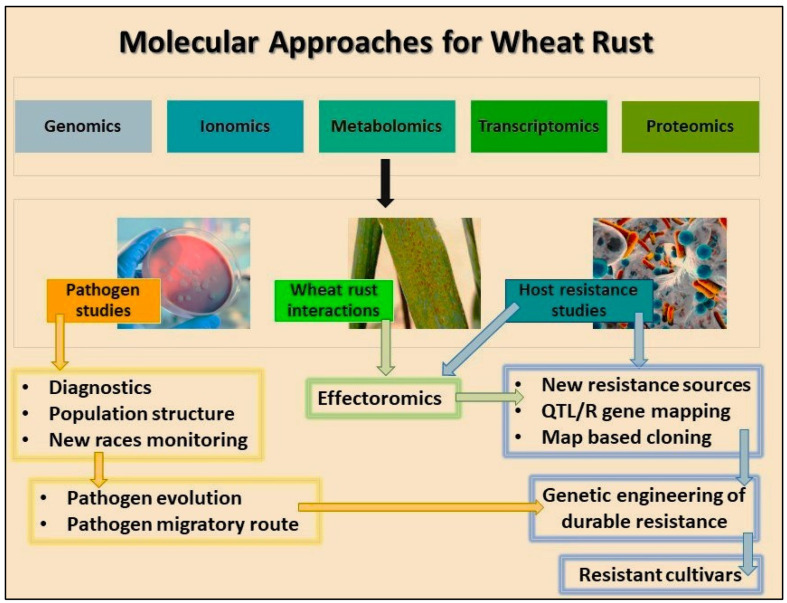
Overview of various molecular techniques and tools used in the study and management of wheat rust, a significant fungal disease affecting wheat crops.

**Figure 2 plants-13-02502-f002:**
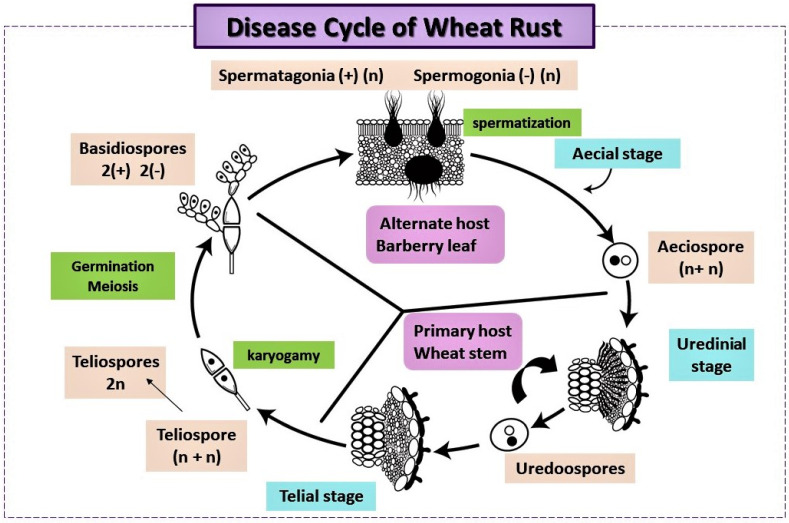
Life cycle of wheat rusts.

**Figure 3 plants-13-02502-f003:**
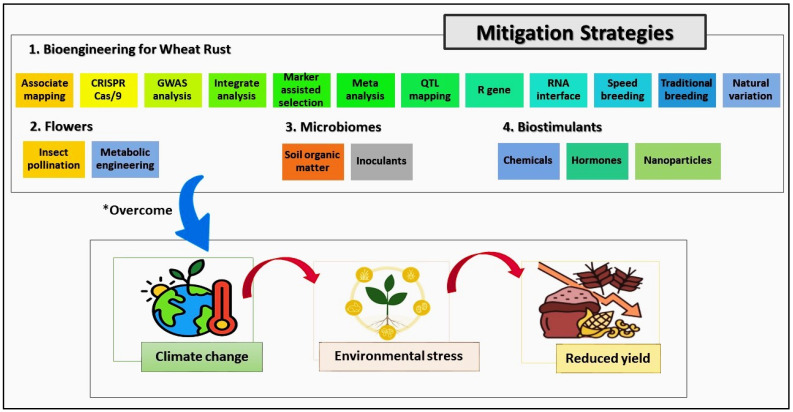
The integration of bioengineering techniques, such as the use of the *Lr34* gene, which provides broad-spectrum resistance to leaf rust, stripe rust, and stem rust, along with enhanced tolerance to environmental stresses like drought and heat. The inclusion of *Lr34* into wheat varieties has demonstrated effectiveness in improving crop resilience under both biotic and abiotic stresses, leading to more stable yields in the face of climate variability. The asterisk (*) indicating three challenges: climate change, environmental stress, and reduced yield. This suggests that the strategies mentioned in the upper section (Bioengineering for Wheat Rust, Flowers, Microbiomes, and Biostimulants) are designed to mitigate or “overcome” these challenges affecting crop production.

**Figure 4 plants-13-02502-f004:**
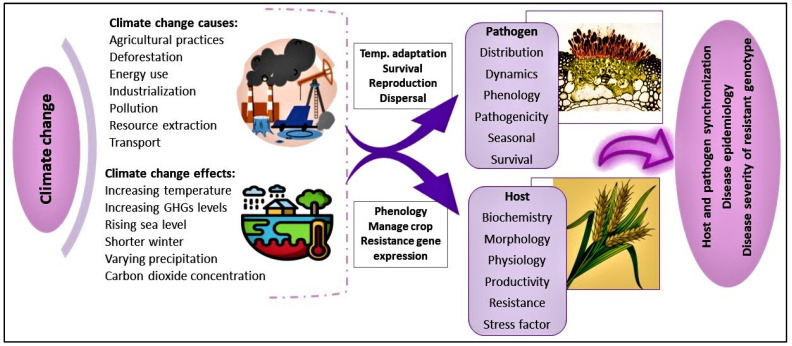
Climate change has a significant impact on the epidemiology of pathogens and host.

**Table 1 plants-13-02502-t001:** Disease spatial distribution map of wheat rusts, highlighting the importance of understanding the spatial distribution and economic significance of wheat rusts to develop targeted and effective control measures, ensuring the stability of wheat production globally.

Wheat Rusts	Current Importance				Historical Importance			
	Major	Minor	Local	Rare	Major	Minor	Local	Rare
Leaf Rust	North Africa Central and Southeast Asia East Europe America	--	East and South Africa East, South, and West Asia Australia New Zealand West Europe	--	North Africa Central, South and Southeast Asia Europe America	--	East and South Africa East, South, and West Asia Australia New Zealand	--
Stem Rust	East Africa	Central, South, and Southeast Asia Europe North America	North and South Africa East and West Asia Australia New Zealand South America	--	Africa East, South, and West Asia Australia New Zealand Europe America	Central and Southeast Asia	--	--
Stripe Rust	East Africa East and West Asia West Europe	--	North and South Africa Central and South Asia Australia New Zealand East Europe America	Southeast Asia	East Africa East and West Asia West Europe	--	North Africa Central and South Asia East Europe America	South Africa Southeast Asia Australia New Zealand

**Table 2 plants-13-02502-t002:** List of cloned genes with wheat rust resistance.

Disease	Resistance Genes	References
Leaf Rust	*Lr1*, *Lr9/Lr58*, *Lr10*, *Lr13*, *Lr14a*, *Lr21*, *Lr22a*, *Lr34*, *Lr42*, *Lr47*, *Lr67*	[51,68,69,70]
Stem Rust	*Sr9*, *Sr13*, *Sr21*, *Sr33*, *Sr35*, *Sr50*, *Sr55*, *Sr57*, *Sr60*, *Sr62*, *Sr22*, *Sr26*, *Sr27*, *Sr45*, *Sr61*, *Sr43*, *Sr46*, *SrTA1662*	[48,49,71,72]
Stripe Rust	*Yr5/YrSP*, *Yr7*, *Yr27*, *Yr15*, *Yr18*, *Yr36*, *Yr46*, *Yr28, YrU1*, *Yr10/YrNAM*	[63,73,74,75,76]

**Table 3 plants-13-02502-t003:** List of common fungicides.

Generic Name	Chemical Name	Refs.
Triadimefon	(RS)-1-(4-chlorophenoxy)-3,3-dimethyl-1-(1H-1,2,4-triazol-1-yl) butan-2-one	[38]
Tebuconazole	(RS)-1-(4-chlorophenyl)-4,4-dimethyl-3-(1,2,4-triazol-1-ylmethyl) pentan-3-ol	[94]
Propiconazole	1-[[2-(2,4-dichlorophenyl)-4-propyl-1,3-dioxolan-2-yl] methyl]-1H-1,2,4-triazole	[95]
Azoxystrobin	methyl (E)-2-{2-[6-(2-cyanophenoxy) pyrimidin-4-yloxy] phenyl}-3-methoxyacrylate	[96]
Epoxiconazole	(2RS,3SR)-1-{(2RS)-2-[4-(2,4-dichlorophenyl) phenyl]-1,3-dioxolan-2-yl] methyl}-1H-1,2,4-triazole	[97]
Cyproconazole	2-(4-chlorophenyl)-3-cyclopropyl-1-(1H-1,2,4-triazol-1-yl) pentan-2-ol	[96]
Flutriafol	(RS)-2,4-difluoro-α-(1H-1,2,4-triazol-1-ylmethyl) benzhydryl alcohol	[98]
Difenoconazole	(E)-1-[2-[2-chloro-4-(4-chlorophenoxy) phenyl]-4-methyl-1,3-dioxolan-2-ylmethyl]-1H-1,2,4-triazole	[99]

## Data Availability

No new data were created or analyzed in this study. Data sharing is not applicable to this article.

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
