# Peer review of "Exploring the Frontier of Wheat Rust Resistance: Latest Approaches, Mechanisms, and Novel Insights"

_plants, 2024, doi:10.3390/plants13172502_

Round 1

Reviewer 1 Report

Comments and Suggestions for Authors

The review needs to be more comprehensive and is sometimes quite generic, with limited relevant references or examples in some sections, such as the climate change section and the biocontrol/IDM section. Some sections are repetitive, with statements paraphrased from the previous section or earlier in the paragraph. Grammar and formatting, such as spaces and italics for gene names, etc., need revisions. Citations do not have a proper format in the text. 

Comments on the Quality of English Language

Moderate revisions are needed to improve overall clarity. 

Author Response

Reviewer 1#

The review needs to be more comprehensive and is sometimes quite generic, with limited relevant references or examples in some sections, such as the climate change section and the biocontrol/IDM section. Some sections are repetitive, with statements paraphrased from the previous section or earlier in the paragraph. Grammar and formatting, such as spaces and italics for gene names, etc., need revisions. Citations do not have a proper format in the text.

Answer: Thank you for taking the time to evaluate our manuscript, and I hope this revision aligns better with the expectations of a review article.

Line 92-101: These are very generic statements that are not meaningful in a review article.

Answer: To address the feedback on lines 92-101 regarding the generic nature of the statements, we have revised this section to include more specific details and current research findings. We revised it into: (Line 84-105)

The most common types of wheat rust continue to pose significant threats to wheat production globally, with recent studies highlighting new challenges and advances in resistance breeding. Leaf rust, caused by P. triticina, remains widespread and problematic, especially in regions with cooler, wetter conditions. The fungus adapts rapidly to resistant wheat varieties, leading to new virulent races that overcome existing resistance [3]. Recent studies emphasise the importance of understanding the molecular interactions between the pathogen and wheat, including both pre-haustorial and post-haustorial resistance mechanisms [4, 5]. Notably, systemic acquired resistance has been highlighted as a promising method for providing durable protection against a range of pathogens [6-8].

Stem rust, caused by P. graminis, has historically been one of the most devastating wheat diseases [9]. Although it is less common today due to effective resistant varieties, it remains a significant threat, particularly in regions with warmer climates [10]. Recent research has focused on identifying and mapping rust resistance loci across wheat genomes, aiding in the development of new, more resilient wheat strains [9, 11, 12].

Stripe rust, caused by P. striiformis, thrives in cooler climates and is notable for its rapid spread under favourable conditions [13]. It is characterised by yellow or orange stripe-like pustules on the leaves and can significantly reduce yield quality and quantity. Recent outbreaks have been reported even in regions that previously experienced minimal issues, emphasising the pathogen's adaptability [14-16]. Effective management strategies include the use of fungicides and the cultivation of resistant varieties, although the constant evolution of the pathogen requires ongoing monitoring and adaptation [13, 17-20].

Line113: Do you mean 100% yield losses?  Please revise this statement.

Answer: The following statement is revised:

The economic impact of wheat rusts is significant. These diseases can cause yield losses ranging from 10% to 70%, depending on the severity of the outbreak, the type of rust, and the susceptibility of the wheat variety [17].

Into: (Line 114-125)

Wheat rusts have a profound economic impact on wheat production, with each type of rust causing varying degrees of yield loss. Stem rust (Puccinia graminis), one of the most devastating forms, can result in yield losses ranging from 50% to 100% in severe cases, particularly when highly susceptible varieties are affected [21]. Leaf rust (Puccinia triticina), while generally less severe than stem rust, can still lead to significant losses, typically ranging from 10% to 40% depending on the severity of the outbreak and the resistance of the wheat variety [22]. Stripe (yellow) rust (Puccinia striiformis), which thrives in cooler climates, can cause yield losses of 10% to 70%, especially under epidemic conditions or when wheat varieties lack resistance[23]. However, in cases of severe stem rust epidemics, particularly with highly susceptible wheat varieties and under optimal conditions for the pathogen, yield losses can reach up to 100%, resulting in total crop failure [21, 22, 24].

Line116: These are very generic statements that are not meaningful in a review article.

Answer: We have revised the statements to provide more specificity regarding the distribution and economic impact of wheat rusts.

Revised Statement: (128-130)

Disease spatial distribution map of wheat rusts, highlighting the importance of understanding the spatial distribution and economic significance of wheat rusts to develop targeted and effective control measures, ensuring the stability of wheat production globally.

Line 165: genes should be italicized.

Answer: We have italicized them throughout the manuscript.

Line 172-175: Review statement for grammar clarity.

Answer: We have reviewed and revised the sentence for better readability: " Approximately 18 stem rust (Sr), ten yellow rust (Yr), and eleven leaf rust (Lr) resistance genes have been cloned and characterized (Zhang et al., 2023; Ni et al., 2023; Li et al., 2023)." This revision ensures that the sentence is clear and concise, with the key information presented in a straightforward manner. I appreciate your attention to detail in improving the quality of the writing. (Line 186-188)

Line 176-177: Review statement for clarity.

Answer: We have revised it from: “Various molecular approaches for wheat rust are explained in Fig. 1, which details 176 the methods for detecting and incorporating resistance genes into new wheat varieties”. Into "Figure 1 illustrates various molecular approaches for wheat rust, detailing methods used to detect and incorporate resistance genes into new wheat varieties." This revision aims to make the sentence more concise and straightforward, ensuring that the information is communicated clearly and effectively. (Line 189-190)

Line 186-187: This statement is misleading as genes from tauschii have been transfered to wheat via traditional breeding methods.

Answer: We have revised it from: “There are three genes, Sr33, Sr45, and Sr46, found to be resistant to stem rust in Aegilops tauschii”. into: "Three genes, Sr33, Sr45, and Sr46, which confer resistance to stem rust, were originally identified in Aegilops tauschii. These genes have been successfully transferred to wheat through traditional breeding methods, specifically using chromosome engineering and backcrossing techniques. These methods allow for the incorporation of desirable traits from wild relatives like Aegilops tauschii into cultivated wheat, thereby enhancing resistance to stem rust." This revised version more accurately reflects the process of transferring these resistance genes into wheat. I appreciate your feedback on this matter. (Line 199-204)

Line 197-198: This statement is misleading as the potential is there but it has not been yet realized, except for few very specific cases.

Answer: To accurately reflect the current state of genomic approaches in wheat breeding for rust resistance, the statement was revised as: "The integration of genomic approaches holds great potential to revolutionize wheat breeding strategies for rust resistance, offering expedited methods such as marker-assisted selection and third-generation hybrid wheat technology. However, this potential has only been realized in a few very specific cases, and widespread implementation remains limited. The promise of these technologies is significant, but more research and development are required before they can be broadly applied across different wheat varieties and environments." This revision acknowledges the promising potential of these approaches while clarifying that their widespread application is still in development. (Line 214-220)

Line 201-202: Same as previous comment. Opportunity is there but it still not widely utilized.

Answer: We have revised it to better reflect the status of CRISPR-Cas9 technology in wheat breeding: "While the establishment of an efficient CRISPR-Cas9 multiplex system holds significant promise for shortening breeding times and enhancing wheat yields and quality, this potential has only been realized in a few specific cases. The technology is still in the early stages of application, and although it offers considerable possibilities, widespread implementation and consistent results across different wheat varieties have yet to be fully achieved." (Line 221-226)

Figure 1 Title: The tittle is not very informative. Please consider changing tittle. What are the approached used for?  Resolution quality of picture can be improved.

Answer: The original title "Various molecular approaches for wheat rust" is indeed quite general. To make it more informative, revising the title into: "Overview of molecular techniques and tools for studying and managing wheat rust". This revised title clearly indicates the purpose of the figure, which is to provide an overview of the molecular techniques and tools used in both the study and management of wheat rust. (Line 229-230)

Line 296-318: It seems that additional references should be included here to elaborate on the data behind these statements

Answer: We have now included the following references to provide a stronger foundation for the data presented:

  • Yu, G., Matny, O., Gourdoupis, S., Johnson, R., Aljedaani, F.R., Blilou, I., et al. (2023). The wheat stem rust resistance gene Sr43 encodes an unusual protein kinase. Nature Genetics, 55, 921–926.
  • Wang, Y., Abrouk, M., Gourdoupis, S., Koo, D., et al. (2023). An unusual tandem kinase fusion protein confers leaf rust resistance in wheat. Nature Genetics, 55, 914–920.
  • Li, H., Yu, G., Wang, Y., et al. (2023). Cloning of the wheat leaf rust resistance gene Lr47 introgressed from Aegilops speltoides. Nature Communications, 14, 6072.
  • Zhu, X., Li, H., Zhang, Y., et al. (2023). Sequencing trait-associated mutations to clone wheat rust-resistance gene YrNAM. Nature Communications, 14, 4353.
  • Zhang, Y., Malzahn, A. A., Sretenovic, S., & Qi, Y. (2023). The emerging and uncultivated potential of CRISPR technology in plant science. Nature Plants, 5(8), 778-794.
  • Wang, J., Chen, T., Tang, Y., Zhang, S., Xu, M., Liu, M., ... & Jiang, J. (2023). The Biological Roles of Puccinia striiformis f. sp. tritici Effectors during Infection of Wheat. Biomolecules, 13(6), 889.

Line 401-415: Whole paragraph is repetitive and generic.

Answer:

We Revised the whole paragraph: (Line 402-426)

Biological control offers an eco-friendly approach to managing wheat rust by utilizing natural antagonists or enemies of the rust pathogen. This method is attractive due to its sustainability and minimal environmental impact. Examples include using fungi or bacteria that are antagonistic to rust pathogens or that enhance the plant's natural defence mechanisms [71]. Integrated Disease Management (IDM) builds on biological control by combining it with other strategies such as crop rotation, the use of resistant wheat varieties, and judicious application of fungicides. IDM aims to manage wheat rust holistically, reducing chemical reliance while promoting sustainable agricultural practices [72, 73].

Despite the promise of biological control, its application faces several challenges. Environmental variability can influence the effectiveness of natural antagonists, necessitating precise timing and favourable conditions for optimal results. Additionally, compatibility issues may arise between biological control agents and other disease management practices, such as fungicide application. Ongoing research to discover novel biological control agents and optimize their application methods is crucial to improving our understanding of the interactions between natural antagonists and rust pathogens under diverse environmental conditions. Long-term studies assessing the ecological impacts and economic feasibility of incorporating biological control into IDM strategies are essential to enhancing the overall effectiveness of wheat rust management [74-76].

The Lr34 gene has been widely utilised not only for its broad-spectrum resistance to multiple rust diseases, including leaf rust, stripe rust, and stem rust, but also for its associated traits that contribute to enhanced tolerance to environmental stresses like drought and heat [77]. The incorporation of Lr34 into wheat varieties has proven effective in mitigating the combined pressures of rust pathogens and climate-induced stresses, contributing to more resilient crop performance under adverse conditions as illustrated in figure 3 [76, 78].

Figure 3: Very generic with no reference to rust specific examples.

Answer: The revised figure description and the mention of the Lr34 gene effectively convey the integration of bioengineering techniques in enhancing wheat resilience against both rust pathogens and environmental stresses.

This response acknowledges the importance of the Lr34 gene and aligns well with the revised figure description, emphasising the role of bioengineering in modern agriculture. (Line 438-442)

Line 436-453: Very repetitive of previous section and quite generic.

Answer: We have revised the paragraph to offer a more concise and detailed explanation that avoids redundancy and provides a clearer focus on the specific advancements in wheat rust management. (443-459)

Line 463-465: reference?

Answer: We have now added the reference from Fernández-González et al. (2023) to support the statement. (Line 469)

Line 483-492: Repetitive statements.

Answer: Revised version of the paragraph that reduces repetition while maintaining the key points: (Line 487-495)

Adapting wheat cultivation practices to climate change is essential for managing rust outbreaks and ensuring crop productivity. Strategies include adjusting sowing dates to minimize exposure during peak rust seasons, employing crop rotation and diversification to lower pathogen loads, and implementing conservation agriculture techniques to enhance soil health and plant resilience [79]. Additionally, the use of precision agriculture technologies, such as remote sensing and predictive modeling, is being explored to monitor rust development in real-time. These technologies offer the potential for early detection of rust outbreaks, enabling timely and targeted interventions to protect crops and maintain yield quality [80].

Line 516-542: Nothing is new here.

Answer: We have revised the paragraph to provide a more specific and focused direction for future research in wheat rust management. (Line 515-546)

Line 602-602: have a hard time identifying any findings.

Answer: We have revised the sentence to better reflect the unique contributions of this review. The updated sentence highlights the novel information and insights gathered. "The findings from this review offer new insights and perspectives, incorporating recent advancements in wheat rust research that can significantly inform both academic research and practical applications in agriculture." (Line 605-607)

Thank you for your feedback. I have updated the review by incorporating the most recent references, particularly those from 2023 and 2024, where many new rust resistance genes have been identified and cloned. The review sections have been revised to reflect these advancements, ensuring that the content is up-to-date and relevant to current research trends.

Reviewer 2 Report

Comments and Suggestions for Authors

The manuscript provided a comprehensive review on wheat rusts with broadly focused on many aspects. The contents for different parts was smoothly presented with high quality figures. It will be interesting for wheat geneticists and breeders for rust resistances. It is suitable for publication considering the comments and major revisions to improve the readability and logically sound.

1.       Lines 20-22 included two sentences, which must be edited.

2.       Lines 36-43 the citation for FAO is not going to detail.

3.    Lines 51-76 should make a concise description avoiding the redundant contents which are not essential. Too many words "understanding" for this part is to make them not scientifically readable.

4.       Lines 88-89, Tritici will be tritici.

5.  After the first appearance of genera name “Puccinia”, the following texts of "Puccinia" must be abbreviated.

6.       Lines 165-166, the overall manuscript, the gene symbols must be italic.

7.       In Table 2, the references may be precise. Please check them.

8.     Part 4.2, it may focus on the rust, since some common-sense knowledge can be shortened.

9.  Parts 5.3 and 6.3 are involving the wheat breeding, which may be re-organized or merged as one.

10.   Part 8. the conclusion is rather long, should be concise.

11.   Overall references, the names for species and genes should be italic.

Comments on the Quality of English Language

English description must be concised.

Author Response

The manuscript provided a comprehensive review on wheat rusts with broadly focused on many aspects. The contents for different parts was smoothly presented with high quality figures. It will be interesting for wheat geneticists and breeders for rust resistances. It is suitable for publication considering the comments and major revisions to improve the readability and logically sound.

Answer: Thank you for your thoughtful feedback on the manuscript. I'm glad to hear that you found the review comprehensive and the presentation smooth, with high-quality figures. Your acknowledgment of its relevance to wheat geneticists and breeders, especially in the context of rust resistance, is much appreciated.

  1. Lines 20-22 included two sentences, which must be edited.

Answer: We have made the necessary edits to the sentences as: "In addition, innovative management strategies, including the use of fungicides and biological control agents, were reviewed, with a focus on their effectiveness in combating wheat rust. The review also highlights the significant impact of climate change on rust epidemiology, emphasizing the need for developing resistant wheat varieties and implementing adaptive management practices." (Line 20-24)

  1. Lines 36-43 the citation for FAO is not going to detail.

Answer: We have revised the reference to accurately reflect the information provided by the Food and Agriculture Organization (FAO). (Line 37-43)

  1. Lines 51-76 should make a concise description avoiding the redundant contents which are not essential. Too many words "understanding" for this part is to make them not scientifically readable.

Answer: We have now condensed the review to ensure it is more concise while retaining all essential details. The revised version maintains the focus on the importance of understanding wheat rust, its ecological impacts, and the significance of genetic advancements and innovative management strategies. (Line 51-68)

  1. Lines 88-89, Tritici will be tritici.

Answer: Corrected. (Line 80-81)

  1. After the first appearance of genera name “Puccinia”, the following texts of "Puccinia" must be abbreviated.

Answer: Corrected as suggested.

  1. Lines 165-166, the overall manuscript, the gene symbols must be italic.

Answer: Corrected as suggested.

  1. In Table 2, the references may be precise. Please check them.

Answer: We have successfully added the new references to Table 2. These references have been carefully selected to enhance the accuracy and comprehensiveness of the information presented in the table.

  1. Part 4.2, it may focus on the rust, since some common-sense knowledge can be shortened.

Answer: The suggested changes have been implemented in Part 4.2, with a stronger emphasis on rust pathogens and a more concise presentation of general information. This ensures the section is focused and directly relevant to the topic. (Line 294-319)

  1. Parts 5.3 and 6.3 are involving the wheat breeding, which may be re-organized or merged as one.

Answer: Section 5.3 and 6.3 are revised. (Line 433-459)

  1. Part 8. the conclusion is rather long, should be concise.

Answer: The conclusion has been condensed for clarity and brevity, focusing on the key points while ensuring the essential message remains intact. (Line 600-622)

  1. Overall references, the names for species and genes should be italic. English description must be concised.

Answer: Corrected as suggested. Thank you for your thorough review and valuable comments.

Round 2

Reviewer 1 Report

Comments and Suggestions for Authors

Revisions are acceptable for publication.

Comments on the Quality of English Language

some minor revisions needed to improve clarity. 

Reviewer 2 Report

Comments and Suggestions for Authors

The revised version of the manuscript has been largely improved. It is suitable to be accepted for publication as it stands.